# Genome-wide analysis of horizontal transfer in non-model wild species from a natural ecosystem reveals new insights into genetic exchange in plants

Emilie Aubin[1]☯, Christel Llauro[1,2]☯, Joseph Garrigue[3], Marie Mirouze [1,4], Olivier Panaud[1,5]\*, Moaine El Baidouri[1,2]\*

1 Laboratoire Génome et Développement des Plantes, Perpignan, Université de Perpignan Via Domitia, Perpignan, France, 2 Laboratoire Génome et Développement des Plantes, Centre National de la Recherche Scientifique, Perpignan, France, 3 Réserve Naturelle Nationale de la forêt de la Massane, France, 4 Diversité, Adaptation, Développement des Plantes, Institut de Recherche pour le Développement, Université de Montpellier, Montpellier, France, 5 Institut Universitaire de France, Paris, France

☯ These authors contributed equally to this work.
\* panaud@univ-perp.fr (OP); moaine.elbaidouri@univ-perp.fr (MEB)

**Data Availability Statement:** The data have been deposited in NCBI under BioProject accession number PRJNA788424 in the NCBI BioProject

## Abstract

Horizontal transfer (HT) refers to the exchange of genetic material between divergent species by mechanisms other than reproduction. In recent years, several studies have demonstrated HTs in eukaryotes, particularly in the context of parasitic relationships and in model species. However, very little is known about HT in natural ecosystems, especially those involving non-parasitic wild species, and the nature of the ecological relationships that promote these HTs. In this work, we conducted a pilot study investigating HTs by sequencing the genomes of 17 wild non-model species from a natural ecosystem, the Massane forest, located in southern France. To this end, we developed a new computational pipeline called INTERCHANGE that is able to characterize HTs at the whole genome level without prior annotation and directly in the raw sequencing reads. Using this pipeline, we identified 12 HT events, half of which occurred between lianas and trees. We found that mainly low copy number LTR-retrotransposons from the *Copia* superfamily were transferred between these wild plant species, especially those of the Ivana and Ale lineages. This study revealed a possible new route for HTs between non-parasitic plants and provides new insights into the genomic characteristics of horizontally transferred DNA in plant genomes.

## Author summary

Horizontal transfer (HT) is known to occur in eukaryotes, but its prevalence among non-parasitic wild species is not well understood. Through genomic analysis of wild non-model species in the Massane forest, we identified 12 HT events primarily involving low copy number LTR-retrotransposons, with half occurring between lianas and trees, unveiling a novel route for HT between non-parasitic plants.

database (https://www.ncbi.nlm.nih.gov/bioproject/PRJNA788424). INTERCHANGE is open source and available at https://github.com/emaubin/INTERCHANGE.

**Funding:** MEB is supported by a grant from the Agence Nationale de la Recherche (ANR-21-CE02-0031-01) and by the CNRS "diversity of biological mechanisms" call (sequencing and data storage). OP is backed by a BQR grant of the University of Perpignan, the Institut Universitaire de France and Occitanie region (Biodivoc) (data collection and sequencing). EA is supported by a PhD grant from the Occitanie region, from which she receives a salary. MM is supported by a grant from the Agence Nationale de la Recherche (ANR-21-PRCI-CE02) (Nanopore sequencing). This study is set within the framework of the "Laboratoire d'Excellence (LABEX)" TULIP (ANR-10-LABX-41) and of the "Ecole Universitaire de Recherche (EUR)" TULIP-GS (ANR-18-EURE-0019). The funders had no role in study design, data collection and analysis, decision to publish, or preparation of the manuscript.

**Competing interests:** The authors have declared that no competing interests exist.

## Introduction

Horizontal transfer (HT) is a process by which genetic material is exchanged between two distinct species without reproduction. HTs are well documented in prokaryotes and considered to play a major role in the adaptation and colonization of new ecological niches [1]. The rapid spread of antibiotic resistance genes among bacteria is a good example of the adaptive role of HTs [2]. Although HTs are thought to be less common in eukaryotes, numerous examples of HTs between multicellular eukaryotes such as plants, animals and insects have been reported in recent years [3–8]. Indeed, over the past decades, the number of sequenced and assembled genomes has steadily increased, facilitating the discovery of several horizontally transferred genes and transposable elements (TEs) between eukaryotes [9–12]. Some of these described HTs were adaptive [13–18]. For instance, in plants, there are several major cases of HTs leading to adaptive innovations such as the recent case of a detoxification gene transmitted horizontally from an endophytic fungus to a wild cereal (*Thinopyrum*) allowing the latter to become resistant to *Fusarium* [17]. This naturally transferred gene was further introduced by breeders into wheat through wide hybridization, resulting in broad resistance to ear blight and crown rot. Another recent example is the acquisition of a detoxification gene through HT in the whitefly, a plant feeding insect, which enables it to overcome host plant defenses [18].

Host-parasite interactions have been shown to promote HTs in eukaryotes, such as in parasitic plants. This is particularly true for the *Orobanchaceae* [19], *Striga* [20], *Cuscuta* [21] and *Rafflesia* [22,23]. In animals, a bloodsucking insect has for instance transferred multiple transposon families between different mammalian species it parasitizes [24]. Natural grafting [25,26] could also facilitate the occurrence of HTs between plant species. There is also strong evidence that HTs occurs between species that do not share any host/parasite interaction [9–12,27].

The vast majority of previous studies on HTs have been conducted using genomic data from public databases of model species for which sympatric relationships and the nature of biological interactions are not always known. This represents a major hurdle in attempts to understand the mechanisms and nature of biotic relationships that can promote HTs in natural ecosystems. Furthermore, while previous reports have shown that both genes and TEs can be horizontally transferred between eukaryotes such as plants, it is not clear whether these two genomic components transfer at the same rate or whether certain types of genes or TEs are more frequently transferred than others. This is because these studies have focused on a group of specific genes or TEs, mainly because the methodologies used so far require prior annotation of the sequences of interest (genes or TEs) limiting any investigation of HTs at the whole genome level.

To address these questions, we conducted a pilot study aiming to investigate HTs in wild plant species from a natural ecosystem, the Massane beech forest located in southern France, considered as one of the last relict forests of the Quaternary Period in Europe [28]. We sampled 17 wild non-model species from this reservation, including trees, climbing plants, herbaceous species and fungi. Through de novo whole-genome sequencing of these species using *Illumina* and the development of a new computational pipeline named INTERCHANGE (for horIzoNtal TransfER CHAracterization in Non-assembled Genome) 12 HTs involving 8 species have been identified. These HTs involve TEs, specifically low copy number LTR-retrotransposons from the *Copia* superfamily. Furthermore, we found that some climbing plants underwent multiple HT events with tree species which could constitute a new route of HT between non-parasitic plants.

## Results

### INTERCHANGE: A new strategy for horizontal transfer identification at the whole-genome scale using short read sequencing data

The inference of HTs is usually based on the use of three criteria [29,30]: i) high sequence similarity between evolutionary divergent species (HS); ii) phylogenetic incongruence between the evolutionary history of the species and that of the transferred sequence (PI); iii) patchy distribution of the transferred sequence in the phylogeny of the species (PD). In other words, no homologous sequence of the transferred DNA is found in the closest relative of the recipient species. There are several alternative hypotheses that could explain high sequence conservation, phylogenetic incongruence and patchy distribution. For instance, high similarity in genetic sequence between divergent species could originate from intense selective pressure or from introgression events. Phylogenetic incongruity could potentially result unrecognized paralogy or incomplete lineage sorting. Finally, patchy distribution might be attributable to the stochastic loss or retention of genes within specific evolutionary lineages. It is critical to underscore that the likelihood of these alternative hypotheses being accurate increase as the species involved in the HT exhibit closer evolutionary distance. Conversely, the integrated application of these criteria, particularly in scenarios involving HTs between evolutionary highly distant species, significantly strengthens the plausibility of HT hypothesis.

Existing methods for detecting HT, which rely on one or two of the previously mentioned three criteria, require prior genome assembly and annotation. This poses a significant challenge to HT studies among wild non-model species lacking accessible references genomes and annotations. We have therefore developed a new pipeline to identify regions of high conservation, potentially indicative of HT events, between two or several genomes. This approach that utilizes raw short-read sequencing data and accounts for all three HT identification criteria. Briefly, INTERCHANGE (i) first identifies similar reads derived from conserved genomic loci between the studied species using a *k*-mer approach (ii) assembles these reads into scaffolds (iii) annotate the scaffolds (iv) and test for high sequence similarity (HS) by comparing the sequence identity between conserved scaffolds with that of orthologous genes. Those HT candidates are then manually tested for the PI and PD criteria. The main steps of this pipeline are shown in Fig 1 and described in details in the method section.

### INTERCHANGE validation using simulated horizontal transfers

To test the accuracy of INTERCHANGE, we simulated HT events involving both genes and transposable elements (TEs) of various classes and superfamilies between three plant models: *Arabidopsis thaliana*, *Oryza sativa* and *Brachypodium distachyon* (Fig 2, see method for details). These species were chosen because they have high-quality, well-annotated reference genomes and present a contrasting evolutionary divergence times. *A. thaliana* and *O. sativa* share a last common ancestor 160 million years ago, while the two Poaceae species, *O sativa* and *B. distachyon*, diverged from each other less than 46 million years ago (http://www.timetree.org/). HTs were simulated between these species with contrasting divergence times in both directions, using each species as a donor and a recipient (Fig 2A). Using the assembled reference genomes, we inserted 100 genes and 100 TEs from *A. thaliana* into the *O. sativa* reference genome, and 100 genes and 100 TEs from *O. sativa* into the *A. thaliana* reference genome, resulting in 400 simulated HT events between these two species (see methods section). Before inserting these sequences into the recipient genomes, random artificial mutations were introduced to simulate both ancient and recent HTs as described in the method section.

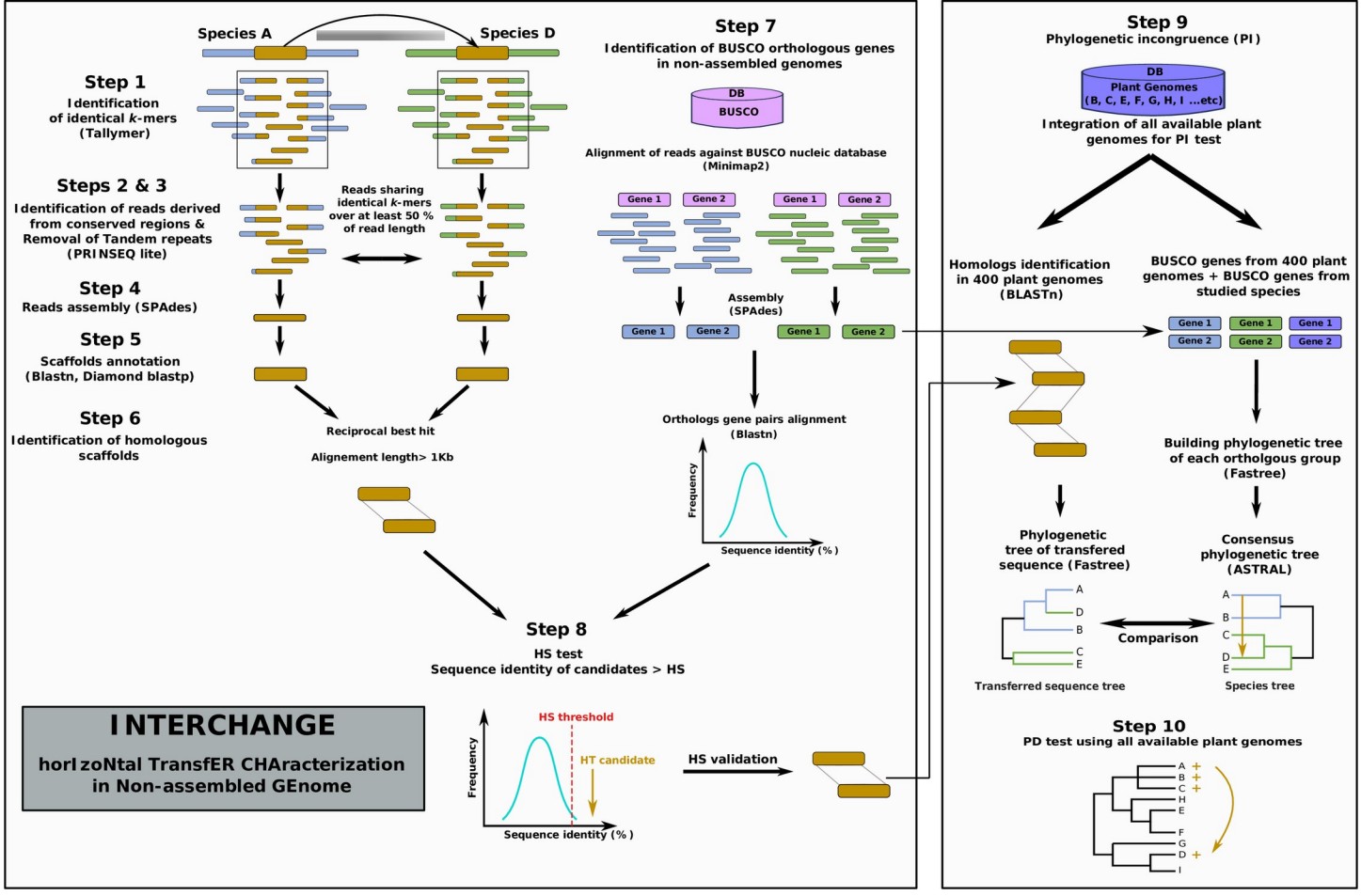

**Fig 1. The different steps of the INTERCHANGE pipeline of horizontal transfer identification from unassembled and unannotated genomes.** Steps 1 to 8 are completely automatic steps 9 and 10 are semi-automatic. Step1: Identification of identical *k*-mers using Tallymer [54]. Steps 2 & 3 Identification of reads derived from conserved regions & Removal of Tandem repeats using PRINSEQ lite [55]. Reads sharing at least 50% of identical *k*-mers are considered as homologous reads. Step 4: homologous reads are extracted and assembled for each pair of species using SPAdes [56]. Step 5: Scaffolds annotation using multiple protein and TEs database: CDDdelta, Repbase, mitochondrial, chloroplast, and ribosomal (TIGR) gene database. Step 6: Identification of homologous scaffolds using reciprocal best hit (RBH). Step 7: Identification of high sequence similarity threshold based on the distribution of orthologous BUSCO gene identities according to the following formula: high similarity threshold (HS) = Q3+(IQR/2); where Q3 is the third quartile, IQR is the interquartile range (Q3-Q1). Step 8: Testing for HS criteria. Step 9: Phylogenetic incongruence criteria. Step 10: testing the Patchy distribution (PD) of transferred sequence. For details see Method section.

Next, *Illumina* short reads were simulated for each genome harboring these *in silico* HTs. The same HT simulation was carried out between *O. sativa* and *B. distachyon*.

The INTERCHANGE pipeline was applied to these species using simulated short read sequencing data from the artificially modified genomes that incorporated simulated HT events. INTERCHANGE was able to identify 90% (361 out of 400) of simulated HT events between *A. thaliana* and *O. sativa* (Fig 2B) (HS = 80%). 10% of simulated HTs were false negatives (39 out of 400) and only one single scaffold candidate was a false positive. Among the identified HTs, 52% were genes and 48% were TEs. For the pair *O. sativa* and *B. distachyon*, INTERCHANGE identified 73% (292/400) of simulated HTs (53% genes and 47% TEs) with a false negative rate of 27% (109 out of 400) and 743 false positives. 85% of these false positives correspond to scaffolds smaller than 500 bp and correspond to highly conserved genes with a sequence similarity higher than the HS threshold (i.e HS = 87%). False negatives correspond to simulated HTs with lower sequence similarity than the threshold. This increase in false

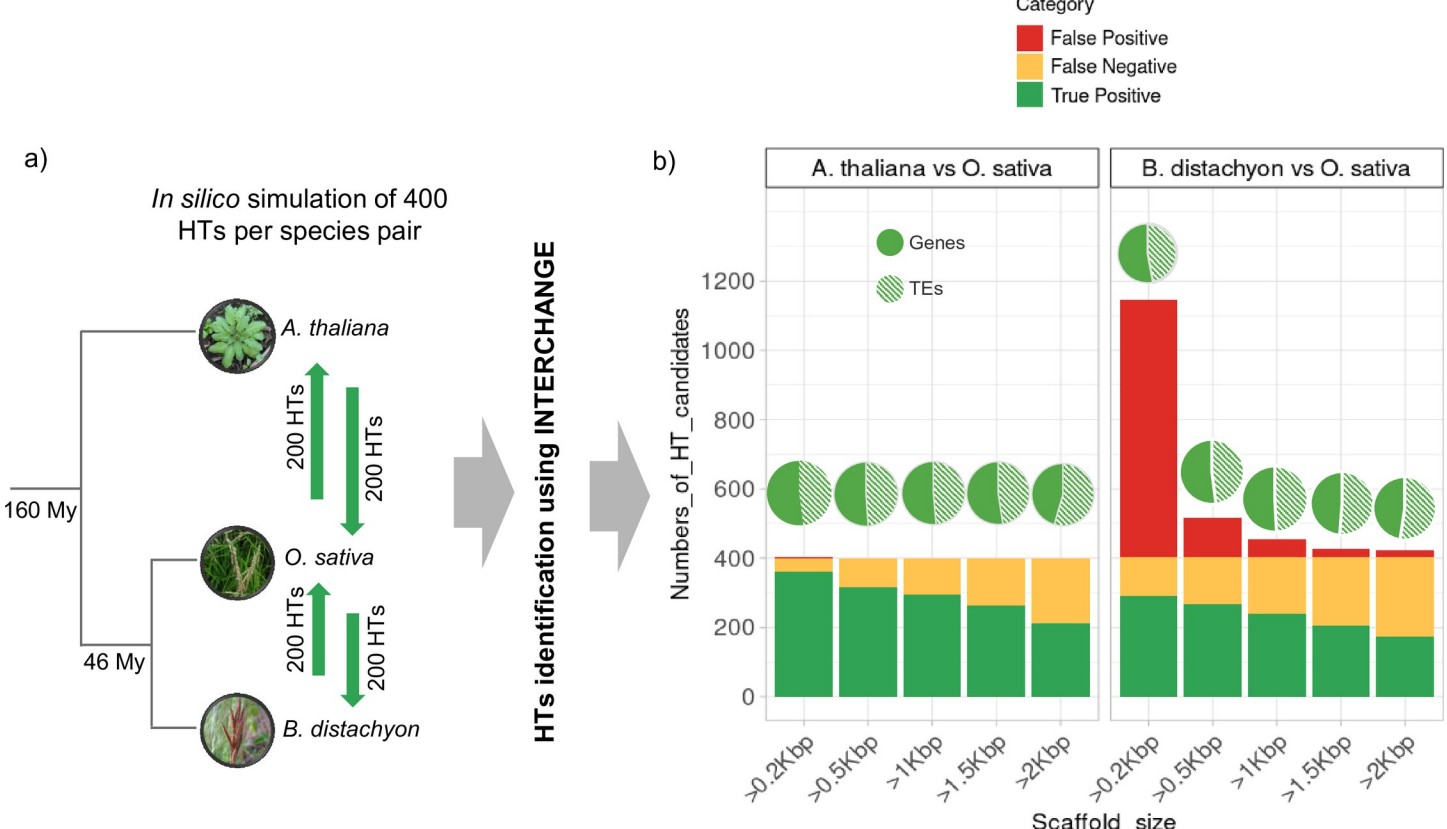

**Fig 2. Simulation of horizontal transfer (HT) between *A. thaliana* and *O. sativa* and between *O. sativa* and B. distachyon.** a. 200 HT events were simulated in each direction (green arrows), comprising genes and TEs with equal proportion. b. INTERCHANGE results using short reads of genomes harboring simulated HTs. Y-axis indicate to the total number of HTs (scaffolds) identified by INTERCHANGE and X-axis represent filters based on scaffold size. The color codes are provided in the figure legend.

negative rate and false positive rate compared to *A. thaliana* and *O. sativa* can be explained by the smaller evolutionary distance between the two Poaceae species. As shown in Fig 2B, when only scaffolds with longer size are considered however, the number of false positives decreases significantly, with a smaller decrease in true positives and without affecting the relative proportion of detected genes and TEs. For instance, by limiting the candidates to those with a scaffold size longer than 1 kbp, the number of false positives decreased by 93% (from 743 to 52) while the number of true positives decreased by only 17% (from 292 to 241) (Fig 2B). This simulation clearly demonstrate that INTERCHANGE is able to efficiently identify simulated HTs directly from the raw reads without any detection bias towards genes or TEs. Additionally, the sensitivity of INTERCHANGE increases as the evolutionary distance between the species involved in HTs increases.

## INTERCHANGE validation using real data allows detection of new horizontal transfer events

To further validate INTERCHANGE pipeline, we applied it to five distant plant genomes for which several HT have been previously reported [9]: grapevine (*Vitis vinifera*), peach (*Prunus persica*), poplar (*Populus trichocarpa*), date palm (*Phoenix dactylifera*) and clementine (*Citrus clementina*). These highly divergent species have experienced 6 HTs of LTR-retrotransposons

named BO1, BO2, BO3, BO4, BO7 and BC1 (BO: HT between plant orders; BC: HT between plant classes) [9]. In this previous study, the identification of these HTs was done through a comparative genomic analysis using assembled and previously annotated genomes. Here, we used the unassembled short reads of the same species (S1 Table) to test whether INTER-CHANGE could detect the previously reported HTs using a minimum scaffold size filter of 1 kbp to reduce the number of false positives as suggested by our HT simulation.

A total of 10 whole genome comparisons were performed between the 5 species and 31 HT candidates were identified using INTERCHANGE, of which 30 correspond to LTR retrotransposons (29 *Copia* and 1 *Gypsy*) and one single gene (*Elongation factor 1*) (S2 Table). In addition to the HS criteria used by INTERCHANGE, we tested for the PI and PD criteria (see Method). The PD criterion was only tested if the HS and PI criteria were satisfied. Nine HT candidates met both the HS, PI and PD criteria, while for the other 22, only one criterion was met leading to their rejection for further analysis (S2 Table). Four among the six known HTs (BO1, BO3, BO4 and BO6) were identified by our new strategy (see Fig 3). BO2 and BC1 were not detected by INTERCHANGE because they did not pass the 1kbp scaffold size filter (S2 Table). Remarkably, INTERCHANGE detected an additional five HTs that were not previously identified by El Baidouri et al. (2014). This includes two HTs between grapevine and date palm (HT1, HT2), one HT between grapevine and poplar (HT3) and two HTs between poplar and peach (HT4, HT5). Interestingly, as with the previously identified HTs, these new HTs correspond to LTR-retrotransposons (LTR-RTs) from the *Copia* superfamily. Strikingly, the LTR-retrotransposons identified by INTERCHANGE as having been transferred between grapevine and date palm (HT1 and HT2) have a high degree of sequence identity at 91% and 95.5%, respectively, suggesting a more recent transfer than the previously identified transfer (86% for BC1) (S2 Table). All newly identified HTs were also found in the reference genomes of species involved in HTs, providing further evidence of the reliability of the INTER-CHANGE pipeline.

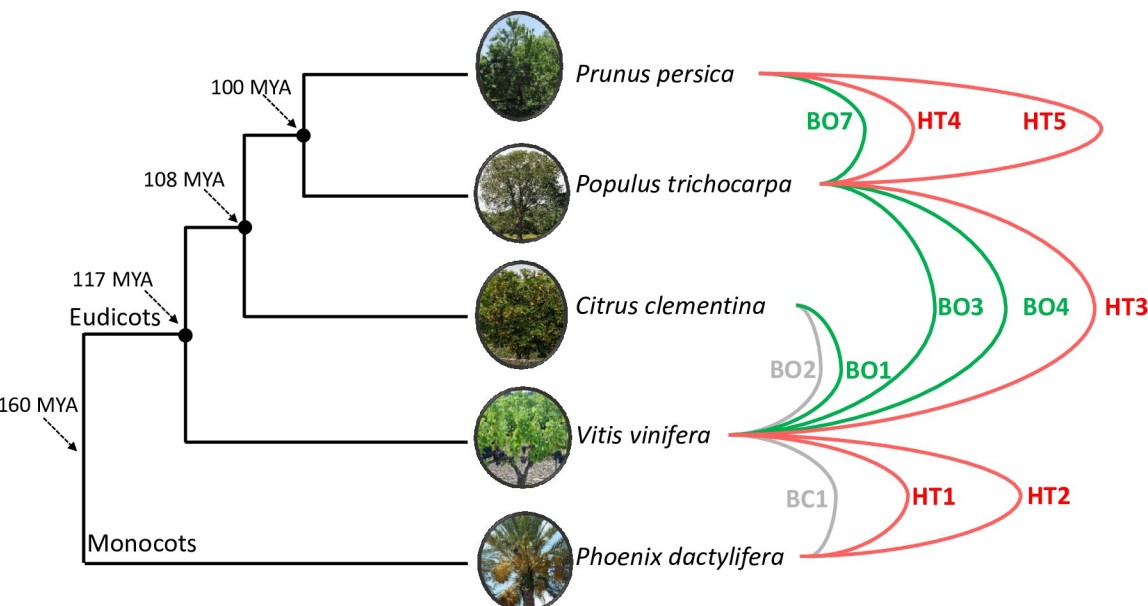

**Fig 3. HTs identified by INTERCHANGE using real data.** Lines represent the HT events identified from genome short read sequencing data. In green, HTs that were identified in a previous study [9] using reference genome and detected by INTERCHANGE from short reads. In gray, HTs missed by INTERCHANGE. In red, new HTs only identified by INTERCHANGE.

To further assess the capability of INTERCHANGE in identifying transferred genes, we focused on the recently reported case of Glutathione S-transferase gene (GST) transfer from the fungi species *Epichloe aotearoae* to the cereal plant *Thinopyrum elongatum* (S1 Table) [17]. As *Illumina* reads for *E. aotearoae* were unavailable in public database, we opted for a closely related species harboring the GST gene and possessing the required data (*Epichloe clarkii*). INTERCHANGE was executed using *Illumina* reads from *T. elongatum* and *E. clarkii*. This analysis successfully detected one significant HT candidate, corresponding to the GST gene homologs of *E. aotearoae* in the *T. elongatum* genome (see S1 Data). Moreover, we performed simulations of *Illumina* reads from *E. aotearoae*, utilizing the *E. aetearoe* reference assembled genome (NCBI: ASM72985v1). A similar analysis was then carried out using INTER-CHANGE, which identified again one candidate that corresponded to the GST gene transfer. Together, this comprehensive evaluation effectively illustrates the capacity of INTERCHANGE to detect both horizontally transferred genes and TEs.

## Characterization of horizontal transfers between wild non-model species from a natural ecosystem

To deepen our understanding of the biotic interactions that can facilitate HTs in natural eco-systems, and to ascertain whether certain genes or TEs are more prone to HT than others, we selected the Massane forest as our study site. This protected reserve, located in southern France, is a unique European location that earned a UNESCO World Heritage Site designation in July 2021. It is mainly composed of beech trees (*Fagus sylvatica*) located between 600 and 1158 m of altitude and covering 336 hectares in total. We sampled 17 different species from this ecosystem including 14 plant species and three fungi (see Table 1). The plant species selected include six tree species, four lianas/climbing plants, two herbaceous, one bramble, and one shrub and are all non-parasitic species. The three fungi species were collected from tree trunks or dead wood and include two saprophytic and one parasitic species. The selection of these species was based on a combination of biological and technical criteria, including: (i) the nature of their biotic interactions, including those with close relationships such as parasitism

**Table 1. Species sampled in the Massane forest and whose genome has been sequenced using *Illumina* short-read sequencing.**

| Common Name | Species | Type | Estimaed Genome size | Available reference genome |
| --- | --- | --- | --- | --- |
| **Beech** | *Fagus sylvatica* | Tree | 540 Mbp | yes |
| **Ash** | *Fraxinus excelsior* | Tree | 840 Mbp | yes |
| **Montpellier maple** | *Acer monspessulanum* | Tree | 730 Mbp | no |
| **Wild cherry** | *Prunus avium* | Tree | 430 Mbp | yes |
| **Alder** | *Alnus glutinosa* | Tree | 500 Mbp | yes |
| **Whitebeam** | *Sorbus aria* | Tree | 1.03 Gbp | no |
| **White Bryony** | *Bryonia dioica* | Liana | 1.6 Gbp | no |
| **Honeysuckle** | *Lonicera periclymenum* | Liana | 2.81 Gbp | no |
| **Ivy** | *Hedera helix* | Liana | 1.5 Gbp | no |
| **Black bryony** | *Dioscorea communis* | Liana | 830 Mbp | no |
| **Giant blackberry** | *Rubus ulmifolius* | Bramble | 450 Mbp | no |
| **Narrow-leaved Ragwort** | *Senecio inaequidens* | Herbaceous | 580 Mbp | no |
| **Sage** | *Salvia sp* | Herbaceous | 760 Mbp | - |
| **Hairy Greenweed** | *Genista pilosa* | Shrub | 1.04 Gbp | no |
| **Tinder Bracket** | *Fomes fomentarius* | Fungi | 50 Mbp | yes |
| **Coral Tooth** | *Hericium clathroides* | Fungi | 40 Mbp | no |
| **Oyster mushroom** | *Pleurotus ostreatus* | Fungi | 20/50Mbp | yes |

or physical proximity, and those with no known close interactions; (ii) genome size, with the aim of obtaining sufficient sequencing coverage for the detection of HTs. The selected species have a genome size smaller than 3 Gbp; (iii) abundance in the Massane forest, to facilitate collection; and (iv) taxonomic diversity to optimize phylogenetic representation.

The genomes of the selected species were sequenced using *Illumina* short-read technology with 20X coverage (see Method and S3 Table). Using INTERCHANGE, we performed 136 whole-genome pairwise comparisons to identify highly similar regions between these species that may have originated from HTs. INTERCHANGE detected 68 HT candidates comprising 46 TEs and 22 genes and involving 8 out of the 17 studied species (see S4 Table). In order to avoid redundancy of candidates due to the presence of multiple paralogs, clustering was performed using SiLiX [31], resulting in 48 HT clusters (see S4 Table). To test the PI criteria, HT candidates where aligned to 400 plant genomes using Blastn (See S5 Table). Phylogenetic trees of the transferred TEs were constructed and compared to the phylogenetic trees of species (see Method Step 9). In total, of the 48 HT candidates (22 genes and 25 TEs) that met the HS criteria only 12 TE candidates also met the PI criteria (S2–S13 Figs, S2 Data). 11 of the 12 TE candidates belong to the LTR *Copia* superfamily (named MaCo01 to MaCo12, for Massane *Copia*) and 1 element belongs to the *Gypsy* superfamily (MaGy01, for Massane *Gypsy*). We further checked the presence/absence of these TEs in the genome 400 plant genomes to test for the PD criteria. This analysis clearly showed that these elements have a patchy distribution, thus confirming the occurrence of HT (see S14–S24 Figs). Notably, both the phylogenetic trees and the patchy distribution of these transferred TEs in 400 plant species point to other possible HT of the same elements between multiple plant species (S8 Table). This is further supported by the utilization of RANGER-DTL software [32], which tests for transfer, duplication, and loss scenarios, strongly suggesting that these elements may have undergone multiple horizontal transfers during their evolution (S4 Data).

## Wet-lab validation of the horizontally transferred LTR-retrotransposons

To rule out potential contamination between the investigated species, despite all the precautions taken during sampling (see Methods), we performed PCR and sanger sequencing to check the presence of the identified HTs in the genome of the 8 species involved in HTs. To do this, we re-sampled two additional individuals from each of the 8 species and extracted their DNA. For each transferred LTR-RT, a set of two primer pairs was designed to amplify different regions of each element (S6 Table and S26 and S27 Figs). In the two individuals of the species involved in the HTs, the transferred LTR-RTs were successfully amplified. For one candidate (MaCo11), only one of the two primer sets results in PCR amplicon (S26 Fig). One PCR amplicon from each transferred LTR was selected for sanger sequencing. Multiple sequence alignment of the PCR product sequences and the sequences of the elements detected using INTERCHANGE, as well as the construction of a phylogenetic tree, validate the occurrences of these HTs (S3 Data). We also verified the presence of the transferred LTR-RT in the genomes of the donor/recipient species for which a reference genome is available. These are *A. glutinosa*, *F. sylvatica*, *F. excelsior* and *P. avium*. All the transferred LTR-RTs implicating one of these species where identified in their respective reference genome. Additionally, we sampled and sequenced the genome of two *H. helix* individuals (Ivy A: ∼4Gb; N50 = 14.4 kbp and Ivy B: ∼4Gb; N50 = 14.6 kbp) and one *F. sylvatica* (7.4 Gb; N50 = 20 kbp) using Nanopore sequencing (see Method section). We were also able to unambiguously identify the transferred LTR-RT involving these two species in different Nanopore reads corresponding to different paralogs (see S28 and S29 Figs). Taken together, these results clearly refute the possibility that the HTs identified in this study are the result of contamination.

## Lianas are hotspots of horizontal transfers

Among the 12 HTs that we identified, none involved a saprophytic or parasitic fungi. As shown in Fig 3, these HTs occurred between 8 out of the 17 studied species. The species involved in these transfers are essentially trees and climbing plants. Indeed, 5 of the 6 analyzed tree species have experienced at least one HT event. These are, in decreasing order of HT frequency: *Fraxinus excelsior* (6 HTs), *Fagus sylvatica* (5 HTs), *Alnus glutinosa* (2 HTs), *Acer monspessulanum* (2 HTs), *Prunus avium* (1 HT). For the climbers, two species among the five analyzed have undergone HTs, namely *Dioscorea communis* (5 HTs) and *Hedera helix* (2 HTs). These HTs were identified between phylogenetically distant species that do not belong to the same plant class. In particular, the five HTs involving *D. communis* (Fig 4), which is a monocot species, occurred with eudicot species that diverged over 150 million years ago. Interestingly, most HTs involving *D. communis* (4/5 HTs) and *H. helix* (2/2 HTs) occurred with tree species which may suggest that the close physical relationship between lianas and trees may be a facilitator of HTs between those plants. Additionally, we found that some species pairs underwent multiple independent HTs of different LTR families such as the ones that occurred between *D. communis* and *F. excelsior* (2 HTs) and between *F. excelsior and F. sylvatica* (2 HTs). The direction of the HTs could not be determined, although the patchy distribution is clearly shown for all transferred LTR-RTs (S14–S25 Figs), as this will require further sampling and sequencing of additional plant genomes.

Sequence identity between the transferred LTR-RTs varies from 89 to 97% (S4 Table), corresponding to an age of transfer between 3.8 and 1.15 million years (Mya) (using the molecular clock rate of Ma and Bennetzen, 2004) [33]. This indicates that these HTs occurred millions years ago and are therefore ancients. This is also supported by the PCR analysis on different individuals that indicate that these HTs are likely to be fixed in populations of these species. Additionally, as shown earlier, for the species for which the reference genome is available (*F. sylvatica*, *F. excelsior*, *P. avium* and *A. glutinosa*) the transferred elements were found in their respective genomes pointing to ancient HT events. From the available data, it cannot be determined where these HTs took place and whether they occurred in the Massane forest, despite its ancient origin. It is important to note however that all species involved in these HTs are native to European and Mediterranean regions and that their respective geographic distributions overlap, indicating that they have been in contact for a long period of time, thus facilitating the occurrence of HTs.

## *Copia* LTR-retrotransposons are the most frequently horizontally transferred elements in the investigated plant species

Despite the fact that our approach of HT identification does not focus on specific types of sequences such as TEs or genes, unlike all other approaches, the HTs identified in the Massane forest involve only LTR-retrotransposons. Some might argue that INTERCHANGE has a bias toward LTR-retrotransposons transfer detection, but our simulations have shown that this is not the case. Further characterization of the protein-coding genes of these transferred LTR-retrotransposons shows that 11 out of 12 belong to the *Copia* lineages (i.e. MaCo01 to MaCo11) and one belongs to the *Gypsy* lineage (MaGy01). This result is consistent with what we observed in our previous work, where *Copia* were more frequently transferred than *Gypsy* (28 *Copia* vs 7 *Gypsy*) [9]. However, in order to ascertain this, it is essential to check whether this is not due to an over-representation of the *Copia* superfamily among other LTR-RTs in the surveyed genomes. For this purpose, we estimated the relative frequencies of *Copia* and *Gypsy* in the 8 species involved in HTs by aligning their raw genomic reads to a collection of reference protein sequences [34] using Diamond Blastx [35] (See Method). As shown in Fig 5A, *Copia* elements were more prevalent than *Gypsy* elements in 6 out of the 8 species, equally abundant in *D. communis* and less prevalent

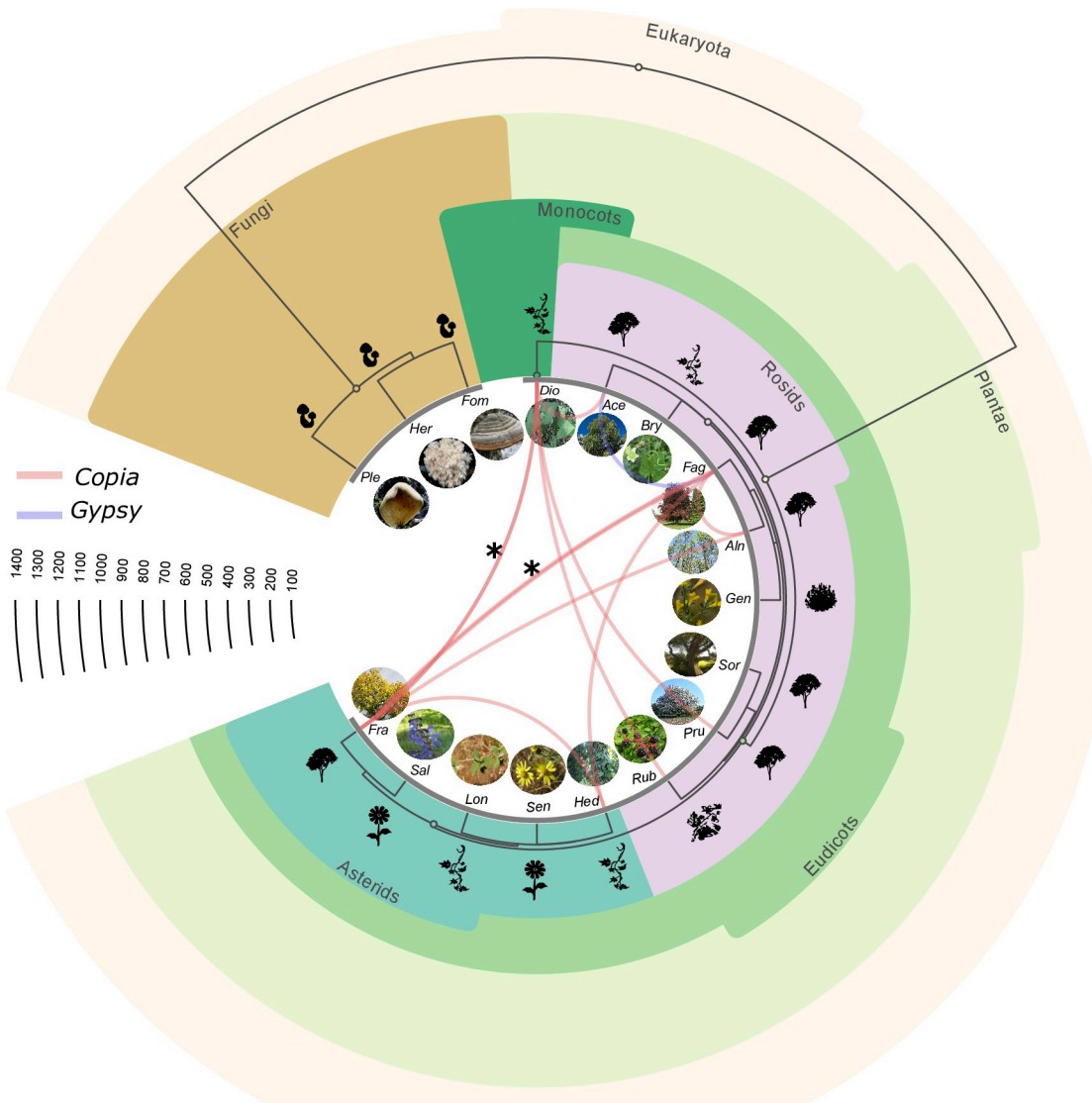

**Fig 4. The phylogenetic tree of the 17 analyzed Massane species.** The curves represent the identified HTs and link the involved species. Blue and red curves represent *Gypsy* and *Copia* HTs, respectively. The asterisks indicate multiple HTs. The horizontal scale represents the divergence time in million years (source: timetree.org). The figure was generated using scripts from Zhang et al. [12]. Correspondence of species names: Ace: *Acer monspessulanum*, Aln: *Alnus glutinosa*, Bry: *Bryonia dioica*, Dio: *Dioscorea communis*, Fag: *Fagus sylvatica*, Fra: *Fraxinus excelsior*, Fom: *Fomes fomentarius*, Gen: *Genista pilosa*, Hed: *Hedera helix*, Her: *Hericium clathroides*, Lon: *Lonicera periclymenum*, Ple: *Pleurotus ostreatus*, Pru: *Prunus avium*, Rub: *Rubus ulmifolius*, Sal: *Salvia sp*, Sen: *Senecio inaequidens*, Sor: *Sorbus aria*.

in *H. helix*. On average, *Copia* were 1.4 times more prevalent than *Gypsy*. However, this cannot explain that 11 out of the 12 identified HTs belong to the *Copia* clade.

## *The* transferred *Copia* LTR-retrotransposons belongs to Ale and Ivana lineages

We then investigated whether some *Copia* lineages have a greater propensity to transfer than others. To this end, we extracted from the Rexdb database the reverse transcriptase (RT) protein sequences of 17 *Copia* reference clades described in the literature [34] as well as those of

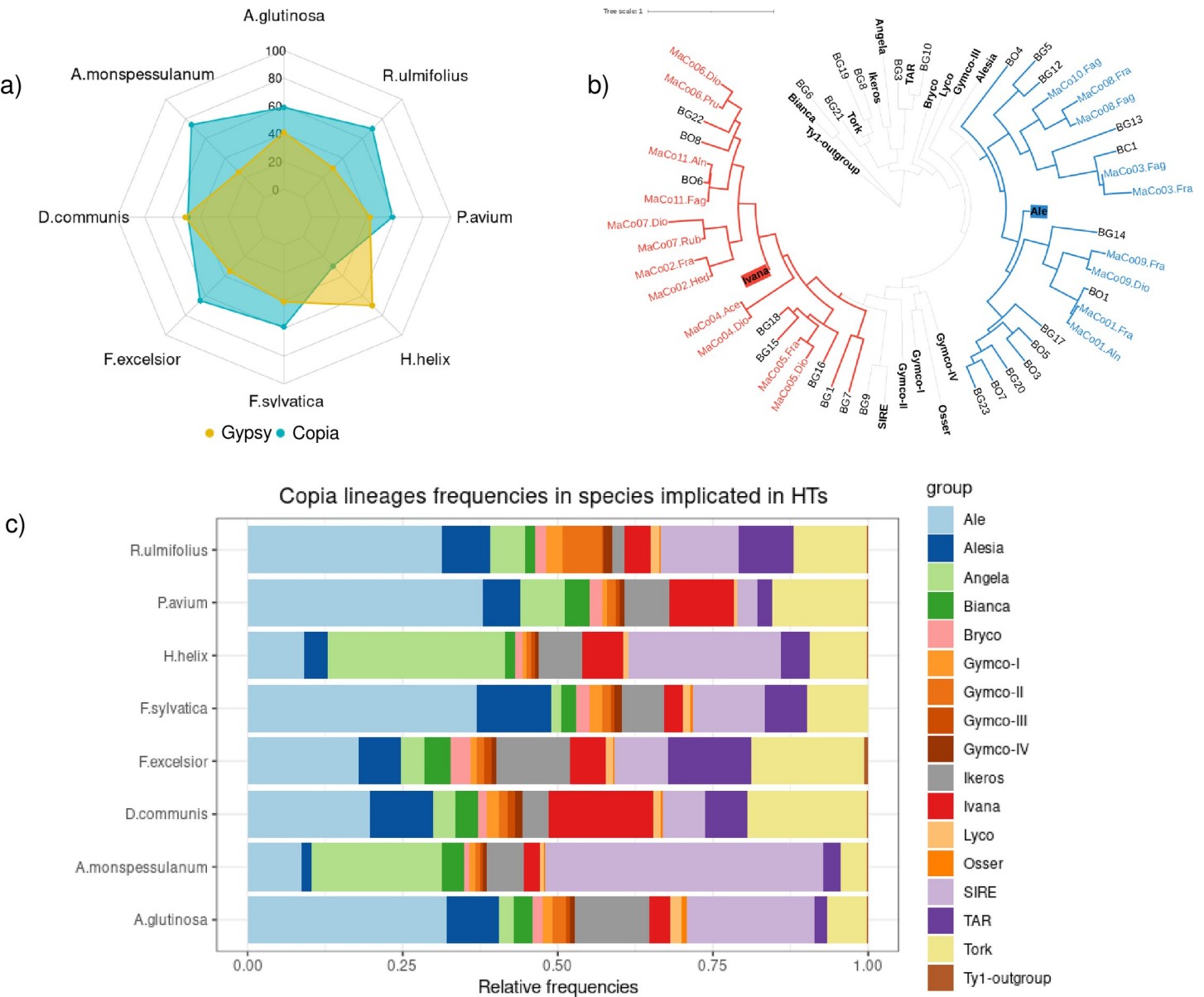

**Fig 5. The relative abundance of LTR-retrotransposon superfamilies in species that have experienced HTs.** a) Relative frequency of *Copia* and *Gypsy* in the studied species involved in HTs. In blue: *Copia* frequency, in yellow: *Gypsy* frequency b) Phylogenetic tree of transferred *Copia* detected in this study using the RT domain. In bold, the consensus sequence of the reference *Copia* lineages. Maco1 to 11 correspond to horizontally transferred elements identified between the plant species from the Massane. BO1 to BO8, BG1 to BG and BC1 correspond to *Copia* elements identified in our previous study. Correspondence of species names as in Fig 3C) *Copia* lineages relative frequencies in species involved in HT.

the 11 transferred *Copia* (MaCo01 to MaCo11) identified in this study (21/22 paralogs) (see Method). Furthermore, we also extracted the RT domain of the 28 *Copia* previously identified as horizontally transferred between several plant species [9]. The constructed phylogenetic tree shows that for the previously described HTs, 21 *Copia* (75%) belong to only two lineages Ale (13) and Ivana (8) where the 7 remaining HTs belong to different lineages such as Tork, TAR or Bianca (Fig 5B). This trend was even more pronounced for the *Copia* elements identified in the Massane forest. In fact, all transferred elements belong only to these two lineages: Ivana (6 /11) and Ale (5/11) (Fig 5B).

These results suggest that these two lineages are more prone to HTs compared to other *Copia* lineages in the studied species. In order to check whether this observation could be due to an overrepresentation of these two *Copia* lineages in the analyzed plant genomes, we estimated the frequency of all known *Copia* clades in the 8 plant species involved in the identified HTs (see Method). This analysis shows big disparities in *Copia* lineage frequencies in those species with no particular conserved trend. In five of the eight species involved in HTs (*A. glutinosa*, *D. communis*, *F. sylvatica*, *P. avium*, *R. ulmifolius*) the Ale lineage was the predominant *Copia* lineage ranging from 8.6% in *A. monspessulanum* to 37.8% in *P. avium* (Fig 5B). On average, Ale accounted for 24% of the *Copia* elements in these genomes, followed by SIRE lineage (16.6%). Meanwhile, the percentage of the Ivana lineage varies from 3.3% in *A. monspessulanum* to 17% in *D. communis* with an average of around 6.5%. These results show that the strong bias observed in transferred elements belonging to Ale and Ivana lineages cannot be explained by the relative abundance of these lineages in the genomes of the species involved in the HTs.

## The horizontally transferred *Copia* are active after their transfer but show low transpositional activity in both donor and receiver species

To better understand the dynamics of the transferred elements, we estimated their copy number in both donor and receiver species using unassembled raw genome sequencing reads. To this end, we used the coverage of single-copy BUSCO genes as a standard to normalize the observed coverage of each transferred element (see Methods). This analysis show that copy number of the transferred elements varies from single copy to 28 copies for the *Gypsy* MaGy01 with an average of 4.3 copy per species (see S7 Table). For the HTs where we could identify the direction of HT, we can observe that some *Copia* elements exist as single copies (3/8), indicating that these elements likely didn't transpose after their transfer. The remaining elements show, on the contrary, a transpositional activity in the host recipient genomes that results in several copies. However, this post-transfer transpositional activity appears to be low, with only 2 to 5 copies observed for each transferred LTR-RT. It is interesting to note that this low transpositional activity is also observed in the donor species, suggesting that it is an inherent feature of the transferred *Copia element*.

## Discussion

### INTERCHANGE a new pipeline for HT characterization at the whole-genome scale using raw sequencing reads

In this study we investigated for the first time HTs between wild non-model species within a natural ecosystem. We sequenced the whole genome of 17 species including trees, climbing plants and fungi and characterized HTs directly form raw sequencing reads thanks to INTERCHANGE pipeline. Using this tool, we were able to report new HT events in previously studied species that had not been identified using previous methods [9]. In this study, we demonstrated the utility of INTERCHANGE for genome-wide screening of HT events between non-model species, using both simulated and real datasets. Importantly, our approach obviates the need for costly and time-consuming genome assembly and annotation, which are often major bottlenecks for conducting such studies.

As indicated by our evaluation of INTERCHANGE some of the known or simulated HT even did not pass the 1 kbp scaffold size filter. A decrease in scaffold size may allow characterization of these events, but this will be at the expense of specificity. Other parameters may also impact the specificity or sensitivity of INTERCHANGE. For example, the smaller the *k*-mer

size, the greater the sensitivity and vice versa, but this will lead to an increase in the computational time needed to perform all the possible pairwise comparisons. The different parameters of INTERCHANGE can be modified by the user which allows great flexibility. However, we shall point out that INTERCHANGE can only detect relatively recent HTs because high sequence divergence between older transferred sequences will not satisfy the HS criteria. Despite these limitations, our results show that this pipeline is very efficient at detecting gene and TEs HT events at the whole-genome scale using unassembled sequencing reads and is therefore a tool of choice for future studies of HT in natural ecosystems. INTERCHANGE could also be used to identify conserved sequences such as homologous genes, TEs or other types of sequences from unassembled genomes, which could be very useful for comparative genomics studies.

## No plant-fungus horizontal transfer was identified at the Massane forest

Using the INTERCHANGE pipeline, we were able to identify 12 HTs implicating 8 plant species. We did not identify any transfer involving fungi even though the three studied species are saprophytic or parasitic and known to proliferate on tree trunks or dead wood. It is broadly accepted that close relationships such as endosymbiosis or parasitism are favorable for HTs in eukaryotes [19,21,24,36–38]. The absence of plant-fungus HT in this study may suggest that such events are rare or too old to be detected [39].

## Liana-tree interactions: a possible new route of horizontal transfer between non-parasitic plants?

Our results also show that the two climbing plants, the common ivy (*H. helix*) and black bryony (*D. communis*) have experienced several HTs events predominantly with trees. These findings are in agreement with our previous study that showed a higher frequency of HTs between grapevines and several tree species [9]. To date, no hypothesis has been put forward to explain this higher frequency of HT in grapevines and whether it is due to an inherent genetic trait or to its particular ecological lifestyle. In fact, similarly to common ivy and black bryony, wild grapevine use trees as support for growth which could explain the high HT frequency observed in this species. A recent study on four different and closely related *Vitis* species seems to confirm this trend [40]. Using comparative genomics, the authors identified dozens of HTs between these four closely related vine species and mainly trees belonging to highly divergent taxa, although they did not highlight the greater frequency of HTs between grapevine and trees. In light of our findings, we hypothesize that liana-tree interactions may favor HTs between non-parasitic plants and could be considered as route by which HTs occur frequently in nature. For the three other climbing plant genomes analyzed in this study, we did not find any HT. Therefore, the question of whether some climbing plants are more prone to HTs than others remains unanswered at this point. This needs to be tested on a larger sampling of liana species. It is also possible that the tree-to-tree HTs that we identified between beech (*F. sylvatica*), ash (*F. excelsior*) or alder (*A. glutinosa*) could be mediated by other, yet not sequenced, climbing plant species.

## Horizontal transfers in plants mainly involve low copy number LTR-retrotransposons belonging to the Ivana and Ale lineages of the *Copia* superfamily

Our study reveals that LTR-retrotransposons are the only genetic elements that experienced HTs in the studied plant species, which confirms earlier reports, but remains without mechanistic explanation. In fact, successful HT requires three key steps, namely the "excision" of

genetic material in the form of DNA or RNA molecules from the donor genome, its transport to the recipient species and finally its integration into the target genome. Due to their transposition life style, LTR-retrotransposons are able to generate extrachromosomal double stranded DNA encapsidated in the VLP (Virus Like Particule) and accumulating in the cytoplasm of the cells [41,42]. They also have the ability to integrate into the host genome using the integrase (IN) [43,44]. LTR-retrotransposons may therefore be more likely to achieve successful HT, given their ability to generate double stranded DNA encapsidated in the VLP and because of their ability to integrate the host genome. Although both *Copia* and *Gypsy* elements can produce VLPs, *Copia* appears to be more prone to horizontal transfer than *Gypsy* elements. If *Copia* and *Gypsy* superfamilies differ mainly in the order of the IN and RT domains, there are some genomic and transpositional features specific to each of these superfamilies. For instance, *Copia* elements are abundant in gene-rich euchromatic regions while *Gypsy* elements are mainly located in Heterochromatic and pericentromeric regions [45]. *Copia* are also generally activated in response to environmental stress as it has been shown for many plant species [46]. It is therefore possible that *Copia* elements, because of their presence in transcriptionally active regions of the genome and because of their responsiveness to environmental stresses could facilitate their HT.

Intriguingly, *Copia* elements that have been horizontally transferred between plant genomes belong mainly to the Ivana and Ale lineage and are low copy numbers. The reasons why Ale and Ivana clade appear to be more prone to HT compared to the other *Copia* lineages remain unknown, as there are no well-known common specific features of these two clades that clearly differentiate them from others [47]. A recent population genomics study in *Arabidopsis arenosa* showed that *Copia* elements, particularly Ale and Ivana respond to temperature and irradiance [48]. It is also interesting to note that the TEs shown to be currently active in *A. thaliana* namely *EVADE* [49] and *ONSEN* [50], also belong to the Ale and Ivana clade, respectively, and the latter is active in response to heat stress [50]. As for the transferred *Copia*, these two families also have a low copy number: two copies for *EVADE* and eight copies for *ONSEN*. When a TE family reaches high copy numbers, it tends to be silenced by the production of small interfering RNAs and the epigenetic machinery depositing DNA methylation [51]. The silenced TEs would not be candidates for HT. If this holds true, the question of the presence and survival of *Gypsy* families in eukaryotic genomes remains to be explained by other mechanisms. Considering our study and previous ones on HT in plants, the propensity of *Copia* elements and in particular Ale and Ivana lineages to transfer horizontally cannot be explained. Further studies are needed to elucidate the reasons for the remarkable ability of low copy number *Copia* to transfer horizontally in plants.

## Conclusions

In this work, we conducted a pilot study on HTs *in natura* in a forest ecosystem. For this purpose, we implemented a new comparative genomics pipeline able to identify HTs at the whole genome level directly from raw sequencing reads. We characterized 12 HTs that all correspond to *Copia* LTR-retrotransposons and particularly those belonging to the Ale and Ivana lineages. Our study also shows that some liana species have experienced recurrent HTs with trees that constitute their growth support in nature. This work sheds light on a new route of HTs between non-parasitic plant species and the type of genetic elements most likely to be horizontally transferred in plants.

### Materiel

**Sampling.**    The 17 species analyzed were sampled in the Massane Forest National Nature Reserve. After sampling the target tissues (leaf or sporophore), the samples were first washed

with a detergent solution (Tween 80 at 0.1%) and then rinsed twice successively in a miliQ water solution. The samples were then dried with absorbent paper and stored in liquid nitrogen and then at -80˚C once in the laboratory before DNA extraction.

### *Illumina* genome sequencing

DNA from each sample was extracted using the CTAB2X [52] and the quality of the DNA was estimated by Nanodrop (Thermo Scientific) and Qubit (Invitrogen) quantification. DNA libraries and sequencing was outsourced to Novogene company using the Hiseq 2000 and Novaseq 6000 platforms. Briefly, a total amount of 1μg DNA per sample was used as input material for the DNA libraries. Sequencing libraries were generated using NEBNext DNA Library Prep Kit following manufacturer's recommendations and indices were added to each sample. The genomic DNA was randomly fragmented to a size of 350bp by shearing, then DNA fragments were end polished, A-tailed, and ligated with the NEBNext adapter for *Illumina* sequencing, and further PCR enriched by P5 and indexed P7 oligos. The PCR products were purified (AMPure XP system) and resulted libraries were analyzed for size distribution by Agilent 2100 Bioanalyzer and quantified using real-time PCR. Paired-end sequencing was performed using a coverage of 20X and a read length of 150bp for each sample.

### Nanopore genome sequencing

High-molecular-weight genomic DNA was extracted from 0.41 g to 0.5 g of frozen leaf tissue according to [53] and the Oxford Nanopore Technologies protocol (February 2019). Briefly, after lysis of cell membranes with Carlson buffer, proteins were removed with chloroform. DNA was purified using Qiagen Genomic-tip 100 columns following the manufacturer's instructions. A selection of fragments > 10 Kb was performed using AMPure XP beads. DNA quantification was performed by Nanodrop (Thermo Scientific) and Qubit assays (Invitrogen) and the quality was assessed on a 0.8% agarose gel. We then followed the 1D genomic DNA protocol by ligation with the SQK-LSK109 kit to prepare the 3 libraries using 3 μg, 3.9 μg, and 4.1 μg of DNA (beech, ivy A, and ivy B), respectively. We successively loaded 1.7 μg of library onto a Flowcell R9, 2.6 μg and 2.7 μg of libraries onto two Flowcell R10. We produced 7.4 Gb, and 2 times 4 Gb of fastq pass reads with N50s of 20 kb, 14.4 kb and 14.6 kb, respectively. Bascalling was performed using guppy in the high accuracy (hac) mode (https://nanoporetech. com/nanopore-sequencing-data-analysis).

### PCR and Sanger sequencing

We utilized the Taq DNA Polymerase 2x Master Mix RED to carry out our PCR amplification. The mix was prepared for each sample by adding 7.5 μl of Taq 2x Master Mix, 6.4 μl of H2O, 0.6 μl of the oligo F and R mix (10μM each), and 0.5 μl of DNA at 10ng/μl. The PCR program we employed consisted of a lid temperature of 98˚C, followed by 34 cycles of denaturation at 95˚C for 30 seconds, annealing at 60˚C for 30 seconds, and extension at 72˚C for 2 minutes. This was followed by a final extension at 72˚C for 5 minutes and a cooling step at 4˚C for 10 minutes. Finally, the samples were electrophoresed on a 1% agarose gel containing TAE 0.5X for 25 minutes at 135 volts, along with a 1kb+ size marker, to visualize the PCR products. PCR products have been purified and sent to Eurofinsgenomics for sequencing using the LightRun Tube platform.

### Detection of Horizontal transfer using INTERCHANGE pipeline

**High similarity criteria (Step1 to 8 using INTERCHANGE automatic pipeline).** Step 1 —Identification of homologous reads derived from conserved regions using a *k*-mers

approach: $k$-mers indexes ($k$ = 30) were generated using Tallymer mkindex option [54]. with default parameters except for: -mersize 30; minocc 1. The search for identical $k$-mers between each species pair was performed using Tallymer search option with the following parameters: -output qseqnum qpos counts sequence.

Step 2—Once identical $k$-mers have been identified between reads of two species, the overlapping $k$-mers are merged and the total similarity score is calculated for each pair of reads using the following formula: Read similarity = total length of identical non-overlapping $k$-mers / reads length. Reads with a similarity score greater than 50% are considered to originate from conserved homologous regions and are therefore kept for further analysis.

Step 3—There are a significant number of identical $k$-mers that correspond to regions of simple repeats such as tandem repeats. Reads containing such repeats are removed using Prinseq-lite tool [55] with the following parameters: out_format 1; -lc-method dust; -lc-thresholds 10.

Step 4—The homologous reads that pass the similarity filter are then extracted and assembled separately for each species using SPAdes [56] with the paired-end and only_assemble options. This step will result in the assembly in each species of scaffolds corresponding to highly conserved regions potentially derived from HTs.

Step 5—The assembled scaffolds are then aligned using both Diamond blastp [35] and BLASTn against several databases with a minimum e-value de 1e-5 et 1e-20 respectively: CDDdelta, Repbase, mitochondrial, chloroplast, and ribosomal (TIGR) gene database [57]. First, sequences that align to mitochondrial, chloroplastic and ribosomal genes are excluded. Indeed, these genes are generally highly conserved between distant species and therefore often meet the criterion of high similarity. When a scaffold aligns to several target sequences from multiple databases, only target sequences with the highest alignment score are considered as being homologous. At the end of this step, each scaffold will be classified into one of these categories: genes, TEs, MCRs (mitochondrial, chloroplast or ribosomal genes)

Step 6—Identification of homologous scaffolds: the objective of this step is to identify homologous scaffolds between each pair of compared species. For this purpose, a reciprocal Blastn is performed and homologous scaffolds are identified using the reciprocal best hit method (RBH).

Step 7—In order to distinguish, among the set of conserved scaffolds identified by INTERCHANGE, those that could originate from HT, it is necessary to first test the criterion of high similarity (HS). This means that the similarity of the transferred sequences between the donor and recipient species must be significantly higher than that of orthologous genes. Before assessing this criterion, it is therefore important to identify and assemble the conserved orthologous genes in the investigated species from unassembled short reads.

Characterization of orthologous BUSCO genes from unassembled reads: (i) Since the studied species from the Massane forest did not have any available gene annotation, we have assembled and annotated their BUSCO genes. These genes were used to test the HS criteria and to build the species phylogenetic tree. As a first step, the BUSCO genes of 400 publicly available assembled plant genomes (S5 Table) were identified, resulting in a genomic database of ~169,000 BUSCO genes covering angiosperms, gymnosperms and basal plant species (this database has been deposited on the following link http://gamay.univ-perp.fr/~moaine/Database/). The genomic reads of each sequenced species from the Massane forest were mapped against this BUSCO database by minimap2 [58]. using default parameters. The mapped reads were extracted, merged and assembled by SPAdes [56] using paired-end and -only_assembler options. The resulting scaffolds were then realigned by Blastn against the nucleic BUSCO database and assigned to their corresponding BUSCO genes.

Step 8—Identification of high sequence similarity threshold based on the distribution of orthologous gene identities: In order to identify whether conserved scaffolds have higher sequence similarity compared to orthologous BUSCO genes, a high similarity threshold (HS) is determined based on the distribution of orthologous gene sequence identities according to the following formula: HS = Q3+(IQR/2); where Q3 is the third quartile, IQR is the interquartile range (Q3-Q1).

Step 9—Phylogenetic incongruence criteria (PI)

Building the phylogenetic tree of the studied species: the phylogenetic tree of the studied species is built based on BUSCO genes previously identified in step 7. Multiple alignment of orthologous BUSCO genes of the studied species and the 400 plant genomes is performed using Mafft program [59]. The alignments are then cleaned with TrimAl [60] and the trees constructed with FastTree [61]. A consensus tree is then obtained using Astral [62] from the previously constructed trees.

Building the phylogenetic tree of the transferred sequence: To construct the phylogenetic tree of the transferred elements, we aligned each of these elements to the assembled genomes of 400 plant species using Blastn. Sequences with sequence identity greater than 80% and covering at least 60% of the element were considered homologous. We performed multiple alignments for each element and all its homologs using the Mafft program [59]. These alignments were then cleaned using trimAL [60] with the following parameters: -cons 30; -gt 0.5. Finally, phylogenetic trees for each transferred element were inferred with FastTree [61]. The resulting trees were then manually compared to the species trees to check for the presence or absence of phylogenetic incongruencies. The trees were visualized using the Iroki Phylogenetic Tree Viewer [63]. The PI criterion is met when the phylogenetic tree of the HT candidate shows that the donor and recipient species are sister clades, unlike the species tree.

Step 10- Testing the Patchy distribution (PD): Finally, to consider that there is an unequal distribution of this sequence in the tree of species, the candidate sequence must be found in species close to donor/recipient but missing in species closely related to partner implicated in the HT. Alternatively, the transferred sequence could also be found only in the two species involved in the transfer (S14–S25 Figs).

Candidates meeting the HS, the PI and the PD criteria are therefore considered as resulting from HTs.

## *In silico* simulation of horizontal transfer events

In this simulation, we randomly selected 100 genes and 100 TEs from each species and randomly introduced them into the reference genomes of the other species. For TEs, we selected different copies belonging to the major classes (Class I and Class II) and to various TEs superfamilies (LTRs, LINEs, MuDR, hAT, Mutator, Helitrons, etc.) with an equivalent proportion when possible. Before inserting these genes and TEs into the recipient genome, we artificially introduced mutations to create sequence divergence, simulating both recent and ancient HTs. The mutated sequences had a sequence divergence ranging from 80% to 100% identity compared to the original copies in the donor species. Wgsim tool (https://github.com/lh3/wgsim) was used to simulate 150 bp length paired-end reads with 20X coverage from the donor and recipient genomes carrying the *in silico* HTs using default parameters.

## Estimation of LTR-retrotransposons copy numbers in unassembled genomes

To estimate the copy number of each retrotransposon in the species involved in the transfer, we calculated the number of mapped reads on each transferred retrotransposon compared to

the numbers of mapped reads on single-copy genes. Total reads for each species were mapped onto the transferred LTR-retrotransposons. For each LTR-retrotransposon, we calculated the coverage at each nucleotide of the element. The median coverage was taken as a proxy for the coverage of the element in the genome. The same strategy was adopted to estimate the coverage of the BUSCO genes of the studied species. We then used the following formula to estimate the total copy number of each transferred LTR-RTs using *Illumina* reads: Copy number = (MCT / MCB), where MCT is the median coverage of LTR-RTs and MCB: median coverage of BUSCO genes. To test whether this approach is an appropriate method to estimate copy number using genomic raw reads, we compared the copy number estimated from unassembled genomes with that obtained from assembled reference genomes in species for which the latter is available (S5 Table). Copy numbers estimated from unassembled genomes and those obtained by Blastn against reference genomes are highly correlated, validating our approach (Pearson correlation; R = 0.982, p-value = 4.699E-10).

## Phylogenetic tree of *Copia* lineages

We extract the RT (reverse transcriptase) domain of the transferred *Copia* elements in both donor and receiver species (22 paralogs corresponding to the 11 *Copia* families). For 60% of the paralogs (13/22), the RT domain was assembled using our automatic INTERCHANGE pipeline. For the others, the RT domain was lacking. We then manually reassemble the lacking RT domains using raw *Illumina* reads of the corresponding species. For species for which the reference genome is available (*F. sylvatica*, *F. exclesior*, *P. avium* and *A. glutinosa*), we realigned the raw reads to the reference genomes and used the reference elements as a guide for manual assembly. Alternatively, we used homologs from other closely related plant species to guide manual assembly. Using this strategy, we obtained for most *Copia* paralogs involved in HTs nearly the complete elements with the corresponding RT domain (21/22).

## Frequency estimates of the different *Copia* and *Gypsy* lineages

To estimate the relative frequency of *Copia* and *Gypsy* in the sequenced genomes, we aligned the raw genome reads of each species to a collection of protein sequences corresponding to the different known *Copia* and *Gypsy* lineages from the RexDB [34] database by Diamond Blastx (evalue 1e-5) [35]. The number of aligned reads on each superfamily and on each lineage was reported to the total number of aligned reads to estimate their relative frequency.

## Plant sampling statement

The authors declare that the samples taken in the framework of this study comply with local and national legislation and that an authorization has been granted.

## Supporting information

**S1 Fig. Sequence identity distribution of the assembled BUSCO genes in the studied species.** These sequence identities were obtained by Blastn alignment. N: corresponds to the total number of BUSCO genes that can be aligned at the nucleic level between each pair of species. HS threshold calculated by INTERCHANGE using the following formula: HS = (Q3+IQR/2), the inter-quartile range IQR = Q3-Q1 (Q1 and Q3 correspond to the first and third quartile respectively). The age in millions of years (Mya) represents the divergence time between species according to Timetree.org. Correspondence of species names: Ace: *Acer monspessulanum*, Aln: *Alnus glutinosa*, Bry: *Bryonia dioica*, Dio: *Dioscorea communis*, Fag: *Fagus sylvatica*, Fra: *Fraxinus excelsior*, Fom: *Fomes fomentarius*, Gen: *Genista pilosa*, Hed: *Hedera helix*, Her:

*Hericium clathroides*, Lon: *Lonicera periclymenum*, Ple: *Pleurotus ostreatus*, Pru: Prunus avium, Rub: *Rubus ulmifolius*, Sal: *Salvia sp*, Sen: *Senecio inaequidens*, Sor: *Sorbus aria*.
(PDF)

**S2 Fig. Phylogenetic tree of the *Copia* LTR-retrotransposon MaCo01 constructed using all homologous elements identified in the 400 plant species.** Nodes supported with boostrap values above 70% are indicated with a black dot. Nodes with bootstrap values under 70% are indicated with white dot.
(PDF)

**S3 Fig. Phylogenetic tree of the *Copia* LTR-retrotransposon MaCo02 constructed using all homologous elements identified in the 400 plant species.** Nodes supported with boostrap values above 70% are indicated with a black dot. Nodes with bootstrap values under 70% are indicated with white dot.
(PDF)

**S4 Fig. Phylogenetic tree of the *Copia* LTR-retrotransposon MaCo03 constructed using all homologous elements identified in the 400 plant species.** Nodes supported with boostrap values above 70% are indicated with a black dot. Nodes with bootstrap values under 70% are indicated with white dot.
(PDF)

**S5 Fig. Phylogenetic tree of the *Copia* LTR-retrotransposon MaCo04 constructed using all homologous elements identified in the 400 plant species.** Nodes supported with boostrap values above 70% are indicated with a black dot. Nodes with bootstrap values under 70% are indicated with white dot.
(PDF)

**S6 Fig. Phylogenetic tree of the *Copia* LTR-retrotransposon MaCo05 constructed using all homologous elements identified in the 400 plant species.** Nodes supported with boostrap values above 70% are indicated with a black dot. Nodes with bootstrap values under 70% are indicated with white dot.
(PDF)

**S7 Fig. Phylogenetic tree of the *Copia* LTR-retrotransposon MaCo6 constructed using all homologous elements identified in the 400 plant species.** Nodes supported with boostrap values above 70% are indicated with a black dot. Nodes with bootstrap values under 70% are indicated with white dot.
(PDF)

**S8 Fig. Phylogenetic tree of the *Copia* LTR-retrotransposon MaCo07 constructed using all homologous elements identified in the 400 plant species.** Nodes supported with boostrap values above 70% are indicated with a black dot. Nodes with bootstrap values under 70% are indicated with white dot.
(PDF)

**S9 Fig. Phylogenetic tree of the *Copia* LTR-retrotransposon MaCo08 constructed using all homologous elements identified in the 400 plant species.** Nodes supported with boostrap values above 70% are indicated with a black dot. Nodes with bootstrap values under 70% are indicated with white dot.
(PDF)

**S10 Fig. Phylogenetic tree of the *Copia* LTR-retrotransposon MaCo09 constructed using all homologous elements identified in the 400 plant species.** Nodes supported with boostrap values above 70% are indicated with a black dot. Nodes with bootstrap values under 70% are indicated with white dot.
(PDF)

**S11 Fig. Phylogenetic tree of the *Copia* LTR-retrotransposon MaCo10 constructed using all homologous elements identified in the 400 plant species.** Nodes supported with boostrap values above 70% are indicated with a black dot. Nodes with bootstrap values under 70% are indicated with white dot.
(PDF)

**S12 Fig. Phylogenetic tree of the *Copia* LTR-retrotransposon MaCo11 constructed using all homologous elements identified in the 400 plant species.** Nodes supported with boostrap values above 70% are indicated with a black dot. Nodes with bootstrap values under 70% are indicated with white dot.
(PDF)

**S13 Fig. Phylogenetic tree of the *Gypsy* LTR-retrotransposon MaGy01 constructed using all homologous elements identified in the 400 plant species.** Nodes supported with boostrap values above 70% are indicated with a black dot. Nodes with bootstrap values under 70% are indicated with white dot.
(PDF)

**S14 Fig. Patchy distribution of the horizontally transferred *Copia* LTR-retrotransposon MaCo01 in the phylogenetic tree of 400 plant species.** The green bars represent the species harboring the LTR family and their height the relative abundance in the host genome.
(PDF)

**S15 Fig. Patchy distribution of the horizontally transferred *Copia* LTR-retrotransposon MaCo02 in the phylogenetic tree of 400 plant species.** The green bars represent the species harboring the LTR family and their height the relative abundance in the host genome.
(PDF)

**S16 Fig. Patchy distribution of the horizontally transferred *Copia* LTR-retrotransposon MaCo03 in the phylogenetic tree of 400 plant species.** The green bars represent the species harboring the LTR family and their height the relative abundance in the host genome.
(PDF)

**S17 Fig. Patchy distribution of the horizontally transferred *Copia* LTR-retrotransposon MaCo04 in the phylogenetic tree of 400 plant species.** The green bars represent the species harboring the LTR family and their height the relative abundance in the host genome.
(PDF)

**S18 Fig. Patchy distribution of the horizontally transferred *Copia* LTR-retrotransposon MaCo05 in the phylogenetic tree of 400 plant species.** The green bars represent the species harboring the LTR family and their height the relative abundance in the host genome.
(PDF)

**S19 Fig. Patchy distribution of the horizontally transferred *Copia* LTR-retrotransposon MaCo06 in the phylogenetic tree of 400 plant species.** The green bars represent the species harboring the LTR family and their height the relative abundance in the host genome.
(PDF)

**S20 Fig. Patchy distribution of the horizontally transferred *Copia* LTR-retrotransposon MaCo07 in the phylogenetic tree of 400 plant species.** The green bars represent the species harboring the LTR family and their height the relative abundance in the host genome. (PDF)

**S21 Fig. Patchy distribution of the horizontally transferred *Copia* LTR-retrotransposon MaCo08 in the phylogenetic tree of 400 plant species.** The green bars represent the species harboring the LTR family and their height the relative abundance in the host genome. (PDF)

**S22 Fig. Patchy distribution of the horizontally transferred *Copia* LTR-retrotransposon MaCo09 in the phylogenetic tree of 400 plant species.** The green bars represent the species harboring the LTR family and their height the relative abundance in the host genome. (PDF)

**S23 Fig. Patchy distribution of the horizontally transferred *Copia* LTR-retrotransposon MaCo10 in the phylogenetic tree of 400 plant species.** The green bars represent the species harboring the LTR family and their height the relative abundance in the host genome. (PDF)

**S24 Fig. Patchy distribution of the horizontally transferred *Copia* LTR-retrotransposon MaCo11 in the phylogenetic tree of 400 plant species.** The green bars represent the species harboring the LTR family and their height the relative abundance in the host genome. (PDF)

**S25 Fig. Patchy distribution of the horizontally transferred *Gypsy* LTR-retrotransposon MaGy01 in the phylogenetic tree of 400 plant species.** The green bars represent the species harboring the LTR family and their height the relative abundance in the host genome. (PDF)

**S26 Fig. PCR validation of the transferred LTR-RTs MaCo01 to MaCo06.** Migration direction is from top to bottom. The red arrows indicate the primers designed to amplify different regions of the transferred LTR-RTs in the species involved in the HTs. For each species, PCR was performed using DNA from two different individuals, different from those used for genome sequencing, to limit possible contamination. (PDF)

**S27 Fig. PCR validation of the transferred LTR-RTs MaCo07 to MaCo11 and MaGy01.** Migration direction is from top to bottom. The red arrows indicate the primers designed to amplify different regions of the transferred LTR-RTs in the species involved in the HTs. For each species, PCR was performed using DNA from two different individuals, different from those used for genome sequencing, to limit possible contamination. (PDF)

**S28 Fig. Graphical visualisation of Blastn alignment of transferred LTR-retrotransposon (Maco2) identified and assembled using INTERCHANGE pipeline against Nanopore reads of two *Hedera helix* genomes corresponding to two ivy individuals A and B.** Visual representation was achieved using http://kablammo.wasmuthlab.org/ software. License: https://github.com/jwintersinger/kablammo/blob/master/LICENSE (PDF)

**S29 Fig. Graphical visualisation of Blastn alignment of two transferred LTR-retrotransposons (Maco3 and Maco11) identified and assembled using INTERCHANGE pipeline**

**against Nanopore reads of *Fagus sylvatica* genome.** Visual representation was achieved using http://kablammo.wasmuthlab.org/ software. License: https://github.com/jwintersinger/kablammo/blob/master/LICENSE.
(PDF)

**S1 Table. List of five plant species used as control data and their corresponding sequence read archive IDs used as input in INTERCHANGE.**
(XLSX)

**S2 Table. List of HT candidates detected by INTERCHANGE between the five species listed in S1 Table.**
(XLSX)

**S3 Table. List of species sequenced in the frame of this study and their BioSample IDs.**
(XLSX)

**S4 Table. List of HT candidates detected (MaCo1 to MaCo11 and MaGy01) between the 17 studied species.** In green candidates meeting the HS, PI and PD criteria. In gray candidates meeting only the HS criteria.
(XLSX)

**S5 Table. List of 400 plant species in which homologs of the transferred LTR-RTs were screened.**
(XLSX)

**S6 Table. Primers used for PCR validation of transferred LTR-RTs.** For each candidate, two sets of primers were designed and labeled P1 and P2 (see S26 and S27 Figs).
(XLSX)

**S7 Table. List of the 12 transferred LTR-RTs, their size and copy number estimated from the unassembled genomes and from the reference genome when available.**
(XLSX)

**S8 Table. Multiple horizontal transfer scenarios for LTR-RTs elements MaCo01 to MaCo11 and MaGy01 as identified by the RANGER-DTL Program [32].**
(XLSX)

**S1 Data. Fasta sequence of the scaffold identified by INTERCHANGE, corresponding to the transferred Glutathione S-transferase (GST) gene between Epichloe and T. elongatum, and the reference GST gene.**
(XZ)

**S2 Data. Fasta sequence of the transferred LTRs MaCo01 to MaCo11 and MaGy01.**
(XZ)

**S3 Data. Multifasta of PCR product sequencing and phylogenetic tree of the transferred LTR-RTs MaCo01 to MaCo11 and MaGy01.**
(XZ)

**S4 Data. Annotated phylogenetic trees of MaCo1 to MaCo11 and MaGy01 obtained using RANGER-DTL Program [32] (see S8 Table for correspondance).**
(XZ)

## Acknowledgments

We are grateful to Marie-Christine Carpentier, Joris Bertrand for their help for plant sampling at the Massane forest.

## Author Contributions

**Conceptualization:** Olivier Panaud, Moaine El Baidouri.

**Data curation:** Emilie Aubin, Christel Llauro, Moaine El Baidouri.

**Formal analysis:** Emilie Aubin, Christel Llauro, Moaine El Baidouri.

**Funding acquisition:** Olivier Panaud, Moaine El Baidouri.

**Investigation:** Emilie Aubin, Christel Llauro, Marie Mirouze, Moaine El Baidouri.

**Methodology:** Emilie Aubin, Moaine El Baidouri.

**Project administration:** Olivier Panaud, Moaine El Baidouri.

**Resources:** Emilie Aubin, Christel Llauro, Joseph Garrigue, Olivier Panaud, Moaine El Baidouri.

**Software:** Emilie Aubin.

**Supervision:** Olivier Panaud, Moaine El Baidouri.

**Validation:** Christel Llauro, Moaine El Baidouri.

**Visualization:** Emilie Aubin, Christel Llauro, Moaine El Baidouri.

**Writing – original draft:** Emilie Aubin, Christel Llauro, Moaine El Baidouri.

**Writing – review & editing:** Moaine El Baidouri.

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
