## [Decision Letter · Decision Letter 0]

31 May 2023

Dear Dr El Baidouri,

Thank you very much for submitting your Research Article entitled 'Genome-Wide Analysis of Horizontal Transfer in Non-Model Wild Species from a Natural Ecosystem Reveals New Insights into Genetic Exchange in Plants' to PLOS Genetics.

The manuscript was fully evaluated at the editorial level and by two independent peer reviewers. As you will see from the detailed comments, both read your manuscript very carefully. They appreciate your approach to document horizontal gene transfer without the need for assembled genomes and agree that this is helpful for non-model species. However, there is some doubt, more in review 1, but also in review 2, if the choice of the LTR retrotransposons is optimal to document the HT, as there could be other mechanisms. Reviewer 1 suggests adding a documentation that your protocol confirms a known case of HT for a gene. Further, reviewer 2 requests a more detailed documentation of the pipeline. In addition, both have made a lot of comments how to improve details, writing, and language. Please understand that based on these reviews, we will not be able to accept this version of the manuscript, but we would be willing to review a much-revised version. We cannot, of course, promise publication at that time. If it is not possible to go beyond the LTR retrotransposons, the comments might be useful to prepare the manuscript for an alternative journal, e.g., more specialized on the transposon biology.

If you decide to revise the manuscript for further consideration at PLOS Genetics, please aim to resubmit within the next 60 days, unless it will take extra time to address the concerns of the reviewers, in which case we would appreciate an expected resubmission date by email to plosgenetics@plos.org.

We are sorry that we cannot be more positive about your manuscript at this stage. Please do not hesitate to contact us if you have any concerns or questions.

Yours sincerely,

Ortrun Mittelsten Scheid

Academic Editor

PLOS Genetics

Claudia Köhler

Section Editor

PLOS Genetics

Reviewer's Responses to Questions

**Comments to the Authors:**

Reviewer #1: This is a description of a pilot study designed to test the hypothesis that relatively inexpensive Illumina sequencing can be rapidly detect horizontal transfer events from non-model species. This would facilitate more wide-spread detection of horizontal transfer in natural settings. In theory, surveys of those populations performed for other reasons could be subject to this analysis pipeline. The authors have convincingly demonstrated that they can in fact use this pipeline for that purpose. My concern here is that although this will be of interest for a specialized audience, the results themselves are not significantly different from those already obtained in previous studies. I could have imagined that the same protocol could be used in environment more likely to be illuminating, such as an island environment that has undergone a series of invasions of non-native species from widely divergent places.

Introduction: The authors have provided numerous references supporting the idea that horizontal transfer is common; collectively, the data are quite convincing. However, I would suggest a note of caution. The evidence for HT varies widely depending on the particular study, and the caveats that were once carefully included have been abandoned. Particularly because this is a self-described pilot study, I would have expected a more thorough discussion of potential issues with HT claims upfront, rather than buried in the text (although I do appreciate that the authors do in fact address most potential issues there).

Page 3, line 28. This would be a good time to discuss other possible reasons to reject the HT hypothesis. For instance, high birth death genes, such as R genes and transposable elements (TEs) are prone to exhibit phylogenetic incongruity and patchy distribution.

Page 4, line 19. This is a sensible approach, but why not also use the approach on other, well documented examples of HT? This would be of interest because it would both provide a test of the protocol and, perhaps shed additional light on those HT events. And I see that the authors did just that! However, I would note that although LTR TEs are much more likely to be subject to HT, due to their repetitive nature, high rates of birth-death, and unusual pattern of selection they may not have been the best example. I would think a functional gene that has been unambiguously demonstrated to have been horizontally transferred would have been a better test.

Page 7, line 10. A description of evidence for the historical distribution of these species would have been helpful. Are any of them, for instance, recently introduced?

Supplemental figure 26. These gels of PCR amplicons look unusual. Are they vertical rather than horizontal gels?

Supplemental figure 27. Why do the amplicons of MaCo10 look like the same size? As does MaCo05 in Sup figure 26.

Page 8, line 8. I assume the authors mean “LTR elements” and not just “LTRs”

Page 8, line 22. Was it possible to determine the age of these elements based on LTR similarity? Were the target site duplications identified?

Page 9, line 11. 97% is 1.1 million years?

Page 9, line 28. “LTRs-retrotransposons” should be LTR-retrotransposon

Page 10, line 10. I would imagine that there could be other, technical reasons, for the disparity.

Page 11, line 14. For many reasons, the single element now present may not be at the position of the element when it transferred.

Page 14, line 19. Is it possible that with relatively low sequencing depth, higher copy number elements would be more difficult to identify relatively rare examples of HT is a high copy sea of elements in both the donor and the recipient species.

Reviewer #2: The reviewed manuscript on horizontal transfer "in natura" presents genomic sequencing of 17 wild forest species and a computational pipeline to uncover likely horizontal transfer events in their genome.

The approach is original, more specifically, an improvement on previous procedures used to detect horizontal transfer. The advantage of the presented method is the possibility to use raw sequencing data, as opposed to assembled genomes.

I would like to stress that the described research brings several improvements and novel findings against the current state of affairs:

- the methodologies used so far require prior annotation of the sequences of interest

- the authors identified specific TE families that are more likely to participate in horizontal transfer than other families

- the authors identified lianas and vines as plant species that are more likely to participate in horizontal transfer than other species

Overall, I find the research well planned and carried out, using appropriate arguments and methods. I appreciate the testing on simulated and previously analyzed datasets. However, I identified two problematic issues (labelled as Major Comments below) that should be addressed by the authors. While reading the manuscript I also encountered numerous issues, mostly regarding spelling and grammar which I listed as Minor Comments below.

Major Comments:

1)

The INTERCHANGE pipeline seems to be only introduced by verbal description. In the sake of reproducibility it would be desirable to formalize the pipeline. The minimum acceptable, I think, would be a script with a list of commands and the versions of programs involved. Ideally, however, this would be a formal workflow that could be installed from a repository.

2)

As opposed to genes, in case of LTR-TEs the sequence may exist in many copies. It seems the authors have not accounted for this when carrying out the simulated sequence tests. They also mention several copies identified on page 11.

I am concerned that the two criteria tested in INTERCHANGE, namely phylogenetic congruence and patchiness could be influenced by this. In the case of congruence, could the identified locus be a subsequent transposition of a TE, which also has a congruent copy?

In case of patchiness, could further transposition and repeat deletions contribute to observed patchiness?

Minor Comments:

p2 l1

"especially those of the Ivan and Ale lineages.' <- Ivana?

p2 l28

"Host-parasite interactions has been shown to promote HTs" <- have been

p3 l26

"INTERCHANGE a new strategy for horizontal transfer" <- INTERCHANGE: a new... (or "-")

p4 l3

"require the prior genome assembly" <- required prior genome assembly? -> or maybe better: "Existing methods" instead of "Previous methods", since the sentences continue in present tense. In any case, drop the article "the".

p4 l8

"identifies similar reads derived from conserved locus between the studied species" <- loci (or at least "a conserved locus")

p5 l11

"False negative correspond" <- negatives

p6 l1

"INTERCHNAGE" <- INTERCHANGE

p6 l4

"31 HTs candidates" <- HT candidates

p6 l9

"are satisfied" <- were

"meet" <- met

p6 l14

"INTERCHANGE detect" <- detected

p6 l15

"This include" <- "these include" or "this includes"

p6 l16

"prunus" <- use peach, since English equivalents are used for all other species mentioned in the sentence

p6 l28

"and whether some particular genes" <- and TO FIND OUT whether...

p7 l3

"2 sparophitic" <- saprophytic

p7 l11

"Illumina short-reads technology" <- short read technology

p7 l23 and 24

"belongs to the LTR" <- belong

"Notably, both the

phylogenetic trees and the patchy distribution of these transferred TEs in 400 plant species

point to other possible HT of the same elements between multiple plant species, suggesting

that these elements may have undergone multiple HTs during their evolution." <- I can't help wondering whether too many other species containing the HT sequence, even if in a patchy distribution, does not support an alternative explanation such as occasional loss of a conserved sequence. Also, could the authors name the most important sequences with multiple-species HTs and maybe provide a complete list as supplement?

p9 l21

"their respective geographic distributions overlaps" <- overlap

p10 l11

"belongs to Ale and Ivana lineage" <- belong to Ale and Ivana lineages

p10 l27

"in Copia lineages frequencies" <- lineage

p10 l31

"On average Ale accounted for" <- "On average," <- missing comma

p13

"This needs to be tested on a larger

sampling of liana species. It is also possible that the tree-to-tree HTs that we identified

between beech (F. sylvatica), ash (F. excelsior) or alder (A. glutinosa) could be mediated by

other, yet not sequenced, climbing plant species." <- Perhaps the abilities of root system contacts between trees of different species could also contribute?

p14 l4

"belong mainly to the Ivana and Ale lineage and are low copy numbers" <- belong mainly to the low copy number Ivana and Ale lineages?

p14 l15

"The silenced TEs would not be candidate for HT" <- candidates

p14 l26

"some lianas species" <- some liana species

p14 l31

"Materiel and methods" <- Material

p16 l21

"Once identical k-mer" <- kmers

p16 l22

"overlapping k-mer" <- kmers

p19 l17

"Estimation of copy LTRs number in unassembled genomes" <- Estimation of LTR(-retrotransposon?) copy number(s?) in unassembled genomes

p20 l3

"Phylogenetic tree of copia lineages" <- Copia (capitalize and italicize)

p20

"Multifasta sequences of 12 transferred LTRS (MaCo1 to MaCo12 and MaGy01) are available on this link. PCR product sequences and multiple

alignment of each of the 12 HTs are available on the following link (http://gamay.univ-

perp.fr/~moaine/PCR/)." <- this should be made more permanent, why not include it as a supplement?

**Have all data underlying the figures and results presented in the manuscript been provided?**

Reviewer #1: Yes

Reviewer #2: Yes

PLOS authors have the option to publish the peer review history of their article (what does this mean?). If published, this will include your full peer review and any attached files.

Reviewer #1: No

Reviewer #2: **Yes: **Matej Lexa

---

## [Decision Letter · Decision Letter 1]

11 Sep 2023

Dear Dr EL BAIDOURI,

We are pleased to inform you that the revised version of your manuscript entitled "Genome-Wide Analysis of Horizontal Transfer in Non-Model WildSpecies from a Natural Ecosystem Reveals New Insights into Genetic Exchange in Plants" has been found to address the concern of the previous reviewers and has been editorially accepted for publication in PLOS Genetics. Congratulations!

Yours sincerely,

Ortrun Mittelsten Scheid

Academic Editor

PLOS Genetics

Claudia Köhler

Section Editor

PLOS Genetics

Comments from the reviewers (if applicable):

Reviewer's Responses to Questions

**Comments to the Authors:**

Reviewer #1: The authors have done an excellent job in responding to my concerns.

**Have all data underlying the figures and results presented in the manuscript been provided?**

Reviewer #1: Yes

PLOS authors have the option to publish the peer review history of their article (what does this mean?). If published, this will include your full peer review and any attached files.

Reviewer #1: **Yes: **Damon Lisch

**Data Deposition**

http://datadryad.org/submit?journalID=pgenetics&manu=PGENETICS-D-23-00448R1

**Press Queries**

---

## [Editor Report · Acceptance letter]

28 Sep 2023

PGENETICS-D-23-00448R1 

Genome-Wide Analysis of Horizontal Transfer in Non-Model WildSpecies from a Natural Ecosystem Reveals New Insights into Genetic Exchange in Plants 

Dear Dr EL BAIDOURI, 

We are pleased to inform you that your manuscript entitled "Genome-Wide Analysis of Horizontal Transfer in Non-Model WildSpecies from a Natural Ecosystem Reveals New Insights into Genetic Exchange in Plants" has been formally accepted for publication in PLOS Genetics! Your manuscript is now with our production department and you will be notified of the publication date in due course.

With kind regards,

Zsofi Zombor

PLOS Genetics

On behalf of:
